# Multiparametric Magnetic Resonance Imaging of Penile Cancer: A Pictorial Review

**DOI:** 10.3390/cancers15225324

**Published:** 2023-11-08

**Authors:** Marta D. Switlyk, Andreas Hopland, Edmund Reitan, Shivanthe Sivanesan, Bjørn Brennhovd, Ulrika Axcrona, Knut H. Hole

**Affiliations:** 1Department of Radiology, The Norwegian Radium Hospital, Oslo University Hospital, 0379 Oslo, Norway; edmrei@ous-hf.no (E.R.); khh@ous-hf.no (K.H.H.); 2Department of Urology, The Norwegian Radium Hospital, Oslo University Hospital, 0379 Oslo, Norway; olahop@ous-hf.no (A.H.); shisiv@ous-hf.no (S.S.); bjorb@ous-hf.no (B.B.); 3Institute of Clinical Medicine (KlinMED), Faculty of Medicine, University of Oslo, 0318 Oslo, Norway; 4Department of Pathology, The Norwegian Radium Hospital, Oslo University Hospital, 0379 Oslo, Norway; uaxcrona@ous-hf.no

**Keywords:** dynamic contrast-enhanced MRI, diffusion-weighted imaging, guidelines, multiparametric magnetic resonance imaging, MRI, penile cancer, staging

## Abstract

**Simple Summary:**

Accurate preoperative staging and precise outlining of a tumor’s extent are crucial for selecting the most suitable treatment approach and improving outcomes. The current clinical staging of penile cancer is still largely based on physical examination. Multiparametric magnetic resonance imaging (mpMRI) is an important imaging modality that complements physical examination and reduces uncertainties that can easily arise during this examination. However, evidence for the application of MRI in the assessment of penile cancer is scarce; there is no consensus on MRI protocols, and functional MRI techniques have not been widely studied before. This paper focuses on the diagnostic performance of non-erectile mpMRI in evaluating penile cancer, reviewing the use of functional techniques for comprehensive oncological assessments together with the current literature and the latest guidelines.

**Abstract:**

The role of multiparametric magnetic resonance imaging (mpMRI) in assessing penile cancer is not well defined. However, this modality may be successfully applied for preoperative staging and patient selection; postoperative local and regional surveillance; and assessments of treatment response after oncological therapies. Previous studies have been mostly limited to a few small series evaluating the accuracy of MRI for the preoperative staging of penile cancer. This review discusses the principles of non-erectile mpMRI, including functional techniques and their applications in evaluating the male genital region, along with clinical protocols and technical considerations. The latest clinical classifications and guidelines are reviewed, focusing on imaging recommendations and discussing potential gaps and disadvantages. The development of functional MRI techniques and the extraction of quantitative parameters from these sequences enables the noninvasive assessment of phenotypic and genotypic tumor characteristics. The applications of advanced techniques in penile MRI are yet to be defined. There is a need for prospective trials and feasible multicenter trials due to the rarity of the disease, highlighting the importance of minimum technical requirements for MRI protocols, particularly image resolution, and finally determining the role of mpMRI in the assessment of penile cancer

## 1. Introduction

Primary penile cancer is an uncommon malignancy with an incidence ranging from up to 6.8 per 100,000 in Africa, Asia, and South America to less than 1 per 100,000 in North America and Europe [1,2,3,4]. The age group that is most commonly affected by penile cancer is between 50 and 70 years, although the disease can also be observed in patients under 40 years [5]. In recent years, an increase in incidence has been noted, most likely due to higher rates of human papillomavirus (HPV) infection [6]. Other risk factors include phimosis, poor hygiene, smoking, and chronic inflammatory states [2,7,8]. Squamous cell carcinoma is the most common penile malignancy, most often arising from the glans, sulcus coronarius, or epithelium of the inner prepuce [1,6].

Limited data are available on the role of magnetic resonance imaging (MRI) in assessing penile cancer, with mostly small series evaluating the accuracy of MRI for primary tumor staging [3]. The 2023 edition of the European Association of Urology (EAU)—American Society of Clinical Oncology (ASCO) Guidelines on Penile Cancer provides up-to-date information on the diagnosis and management of penile squamous cell carcinoma [6]. The current clinical staging is still largely based on physical examination, and MRI can be considered in cases that are suspicious for corpus cavernosum infiltration or prior to organ-sparing surgery [6]. Non-erectile multiparametric MRI (mpMRI) provides the precise outlining of tumor extension, showing good concordance with histopathological evaluation for staging, tumor size, and infiltration depth, and should be an important complement to physical examination prior to any decision making [4]. However, the evidence is scarce, there is no consensus concerning MRI protocols, and functional techniques have not been incorporated into the guidelines or widely evaluated. There is a need to establish minimum technical standards for penile MRI that can further be adopted for most major clinical MRI platforms. Our objective with this paper was to discuss and highlight the diagnostic performance of mpMRI in the assessment of penile cancer, and to review the current literature and the latest guidelines proposed for the management of penile cancer. 

## 2. MRI in the Assessment of Penile Cancer

### 2.1. Basic Principles

MRI is the medical application of nuclear magnetic resonance (NMR), based on the interaction of hydrogen nuclei in the presence of an external magnetic field when exposed to radiofrequency waves of a specific resonance frequency [9]. MRI has excellent tissue contrast and does not require the presence of isotopes in the sample or time-consuming calibration, unlike other methods used to characterize the magnetic properties of a tissue, such as the magneto-optic Kerr effect (MOKE) [10], Mössbauer spectroscopy [11], and magnetic force microscopy [12]. 

MRI offers a noninvasive assessment of both primary and locally recurrent penile malignancies [13]. Since its introduction in the early 1980s, MRI has become the reference standard for oncological imaging and has been rapidly evolving, resulting in a variety of sequences, increased quality, and rapid acquisition [14]. Morphological MRI sequences provide excellent soft-tissue contrast and anatomical information. Furthermore, functional MRI sequences provide molecular and physiological information using diffusion-weighted imaging (DWI) and dynamic contrast-enhanced MRI (DCE-MRI) perfusion [15]. Finally, mpMRI allows for the integrated evaluation of morphological images and at least two functional sequences (DWI and DCE-MRI), and has been successfully performed in several malignancies, such as prostate, breast, and bladder cancer [4,16,17,18,19]. Gadolinium-based contrast agents are routinely used in contrast-enhanced MRI, both for dynamic and static applications. The indications and use of gadolinium-based contrast agents are well documented; they are generally safe and well tolerated and have powerful paramagnetic properties [20]. Gadolinium in standard dosages mainly shortens T1 relaxation times, thereby producing a high T1 signal intensity in MRI [21]. 

DWI provides a noninvasive characterization of biological tissues based on the measurements of the random microscopic motion of water protons (Brownian motion). It is performed by using echo planar imaging (EPI) combined with symmetrical motion probing gradients (MPGs) around the 180° refocusing pulse [22,23]. The intensity of MPG pulses is represented by the b-value (s/mm^2^), which measures the strength of the diffusion-sensitizing gradient [22]. Using two b-values, one can map and calculate the apparent diffusion coefficient (ADC) [22]. Malignant tumors typically display low diffusion because of their hypercellularity; consequently, the contrast is accentuated by high signal intensity in the cellular tumor and low signal intensity in the normal background tissue on high b-value images [24]. Low diffusion can also be observed in some benign conditions such as abscesses and hematomas, and DWI findings should not be assessed alone. 

DCE-MRI is another functional technique that is used in oncological imaging. DCE-MRI assesses the passage of blood through tissue vessels, providing information on the density, integrity, and leakiness of the tumor vasculature [25]. It involves sequentially acquired T1-weighted images collected before, during, and after the intravenous injection of gadolinium-based contrast agent. Thus, each acquired image corresponds to one time point and each pixel in each image set gives rise to its own time course, which can then be analyzed with a mathematical model [25,26]. To perform such an analysis, the following measurements are required: (1) the measurement of precontrast T1-values in a tissue, (2) dynamic data acquisition after contrast with reasonably high temporal resolution, and (3) an estimate of the arterial input function (AIF) [25]. DCE-MRI data can be evaluated qualitatively through a visual analysis of the acquired images, semiquantitatively by analyzing the perfusion curve, or quantitatively by analyzing perfusion parameters such as *K^trans^*, *k_ep_*, *v_p_*, and *v_e_* [27]. Tumors can have variable appearances on perfusion; however, high vascular permeability with a rapid rise, strong peak enhancement, and subsequent washout is typical for malignant lesions. 

### 2.2. Clinical Protocols and Technical Considerations

A limited number of reports have assessed the diagnostic potential of MRI in the assessment of penile carcinoma, and inconsistencies in MRI protocols can influence the outcomes and make comparisons difficult [3,4,13,28,29,30,31,32]. Most of these studies did not incorporate modern MRI protocols or anticipated the high spatial resolution of images. Only one study has combined both DCE-MRI and DWI into high-resolution mpMRI, and the initial findings indicated that non-erectile mpMRI can be implemented as a precise diagnostic tool for the preoperative assessment of primary penile carcinoma [4].

Previously, the intracavernosal injection of prostaglandin E_1_ and artificial erection were suggested for MRI. During erection, penile blood flow increases, resulting in an increase in the image size and signal intensity of the corpora cavernosa. This enhances the penile anatomy, particularly the corpora cavernosa and tunica albuginea, and their anatomical boundaries [33]. However, this invasive technique is not routinely used in clinical practice and carries some important contraindications and side effects [28,34]. Furthermore, the pain associated with this procedure may be an important limitation in patients with invasive penile carcinoma, and several patients may experience embarrassment and discomfort during this procedure [4]. The results of a recently published systematic review and meta-analysis showed that MRI with and without artificial erection had similar accuracy for the local staging of penile cancer [28]. According to the latest version of the EAU-ASCO Guidelines for Penile Cancer, artificial erection is not mandatory for MRI [6]. 

A biopsy of penile lesions and histopathological evaluation are important for the diagnosis of penile cancer. Performing MRI prior to biopsy avoids disturbing post-biopsy changes such as hemorrhage and inflammation, which can make the evaluation of smaller, superficial lesions difficult. Moreover, mpMRI can be used to guide biopsies, targeting the most suitable tumor components and avoiding ulcerations or necrosis. 

MpMRI allows for the integrated assessment of tumor morphology, cellularity, and vascularization. Although the data are limited, the use of DCE-MRI and DWI in the evaluation of penile cancers can help to identify small lesions more precisely, allowing to accurately determine the infiltration depth, outline the tumor extension, and detect tumor satellites [4,28,34]. Morphological sequences can over- or underestimate the tumor extent because peritumoral edema and hyperemia are difficult to differentiate from tumor tissue using these sequences [4,35]. Functional sequences seem to depict tumor tissue more selectively and, despite lower spatial resolution, delineate the tumor extent more precisely. DWI visualizes the high cellularity of the tumor as opposed to the surrounding tissue, and the early phases of DCE-MRI visualize the enhancement in the intra-tumoral vessels before the contrast medium reaches the peritumoral vessels [4,24,36]. A summary of our institutional imaging protocol for the assessment of penile carcinoma, which incorporates high-resolution morphological and functional sequences, is presented in Table 1 and Figure 1. The complete protocols for the 1.5-T and 3-T MRI scanners are presented in Appendix A. 

In general, mpMRI of the penis is a challenging examination, where radiologists must evaluate small structures and subtle pathology. Therefore, optimal image quality, an efficient evaluation time, and multidisciplinary cooperation are essential. In our institution, the work is organized within the multidisciplinary penile cancer team and includes the presentation of MRI examinations, photographic tumor images, and histopathology images at multidisciplinary meetings, together with close cooperation with other radiologists within the team as well as urologists, oncologists, pathologists, and nuclear medicine physicians. During the interpretation of the MRI scans, the radiologist plots the tumor position, proximal tumor extent, and satellites on the template drawing to assist pathologists and clinicians in understanding the precise tumor location and surgery planning (Appendix A). 

Various positioning techniques and patient preparation methods have been recommended in the literature. For instance, it has been previously suggested to position the penis dorsiflexed against the lower abdominal wall in the midline and secured in place [3,34]. In our experience, such positioning can cause artifacts due to the motion of the abdominal wall during breathing and decrease the image quality in several patients. Therefore, we prefer to position the penis in a normal anatomical location, pointing downward along the midline. The prepuce is in the normal position and pulled down over the glans. Examination is performed without bowel relaxants or rectal emptying. At our institution, penile MRI is performed without an artificial erection. 

MpMRI is considered safe and well tolerated by most patients, with only a few disadvantages. MRI scans can be contraindicated in a few cases of devices and implants. Moreover, patients with strong claustrophobia may be unable to complete the examination. However, using wide-bore MRI systems, good patient information and sedation when needed can address this issue. Some patients may not cooperate and will not be able to lie motionless during the MRI scan. However, the functional sequences are frequently robust to motion artifacts. Finally, group II gadolinium-based contrast agents (Clariscan) in standard dosages are considered very safe, although caution should be exercised when administering to patients with severe renal impairment [37]. In summary, non-erectile mpMRI is a noninvasive examination performed without artificial erection (without intracavernosal administration of prostaglandin E_1_) and is well tolerated by the majority of patients in different age groups and with varying health statuses. In most cases, mpMRI provides sufficient clinical information, even when part of the examination is hampered by artifacts.

We perform the examinations primarily on a 3-T MRI scanner in order to achieve a high image resolution and signal-to-noise ratio (SNR) within a reasonable scan time. As we need to evaluate small anatomical structures, high-resolution morphological and functional images are crucial. Morphological imaging of the primary tumor and surrounding structures is provided by T2-weighted sequences with a small field of view (FOV), small pixels, and thin, continuous slices, performed in three orthogonal planes. 

Recently, MRI vendors have developed deep learning (DL) techniques to remove noise from the images, such as Deep Resolve Boost delivered by Siemens and AIR Recon DL delivered by GE Healthcare. These techniques improve image quality, reduce the noise, and shorten acquisition time [38,39]. Applying DL enables comparable and high image quality within a reasonable scan time in 1.5-T MRI scanners. Examples of mpMRI protocols are presented in Appendix A, with two protocols for the 3-T MRI scanner (with and without DL), and one protocol for the 1.5-T MRI scanner (with DL). 

DWI of the penis can be challenging because of geometric distortion and susceptibility artifacts from the inhomogeneous magnetic field due to air–tissue boundaries frequently affecting this region. In addition, high-resolution images are difficult to achieve with standard single-shot EPI DWI. T2* decay and EPI blurring (T2* blurring) limit the image quality of the standard single-shot EPI DWI techniques [40]. 

Several DWI techniques have been developed to reduce distortion caused by magnetic susceptibility. These include reduced FOV techniques utilizing spatially selective excitation, such as FOCUS delivered by GE Healthcare, ZOOMit delivered by Siemens, and ZOOM delivered by Philips [41]; readout-segmented multi-shot EPI, such as RESOLVE delivered by Siemens [42]; and phase-segmented multi-shot EPI, such as MUSE delivered by GE Healthcare and IRIS delivered by Philips [43,44]. Sequences that combine these techniques and/or DL are available and help to further improve the image quality. 

These advanced EPI techniques minimize susceptibility and blurring artifacts and are particularly useful for the evaluation of smaller lesions and areas where the magnetic field is inhomogeneous, such as the penis. However, it is important to optimize DWI with parallel imaging to shorten the readout window and minimize the T2* blurring, and with a high sampling bandwidth to shorten echo spacing and reduce the image distortion. In our experience, multi-shot EPI techniques seem to be necessary for 3-T MRI scanners, while reduced FOV techniques are sufficient for 1.5-T MRI scanners. High b-value images are useful for lesion detection and characterization; however, they should be calculated rather than acquired using one of the several available extrapolation models, allowing for a reduced scan time and improved image quality, owing to a higher SNR [45,46]. It is important to avoid using too-high b-values when calculating ADC maps to prevent reaching the noise floor, preferably b0 and b800 [47]. 

Finally, contrast-enhanced imaging, both for dynamic and static applications, can be performed using the Dixon technique, which allows for homogeneous, fat-free imaging, even in the presence of magnetic susceptibility [48]. High-resolution DCE-MRI has previously been shown to detect superficial, tumor-like skin lesions as small as 1–2 mm, and can be beneficial in the work-up of penile cancers [49]. Because of the curved anatomy of the glans, DCE-MRI should be as isotropic as possible, allowing for the multiplanar interpretation of the images. An approximately 1 mm isotropic resolution is achievable using slice interpolation within a dynamic timeframe of 10–15 s. 

## 3. Classification and Staging of Penile Cancer

### 3.1. Classification

The latest 2022 World Health Organization (WHO) classification reinforces the 2016 classification, and subclassifies precursor lesions and invasive tumors into HPV-associated and HPV-independent types (Table 2) [1]. 

There are no established prognostic or treatment differences between HPV-associated and HPV-independent lesions; however, some evidence suggests better prognostic and therapeutic outcomes in HPV-associated malignancies [1,50,51,52]. 

The prognostic role of histology was investigated in previous studies, which showed that the basaloid, sarcomatoid, and adenosquamous subtypes correlated with poorly differentiated tumors and deep tissue infiltration, whereas the verrucous, papillary, and condylomatous (warty) subtypes were associated with low-grade tumors and superficial invasion [53,54].

### 3.2. Staging of Penile Cancer

Penile cancer staging models are useful for planning treatment strategies and predicting prognosis [55]. The TNM staging system is mainly based on stratified anatomical routes of penile cancer spread [55]. The eighth edition of the Union for International Cancer Control (UICC)/American Joint Committee on Cancer (AJCC) TNM classification is currently used to stage penile cancer [6,55,56]. This version was last updated in 2017, and one of the major changes concerned the tumor involvement of the corpus cavernosum, corpus spongiosum, and urethra [34,56]. In the eighth edition, T-stage T2 is defined as the involvement of the corpus spongiosum, while the involvement of the corpus cavernosum (including the tunica albuginea) is defined as stage T3. Urethral involvement is no longer relevant to local staging (Table 3 and Figure 2).

Reliable preoperative staging is crucial for decision making and decreasing the recurrence rate after surgery. Penile amputation is associated with significant functional, sexual, and psychological deficits despite high local control rates [57,58]. A majority of early penile carcinomas are amenable to organ-sparing surgery [13], which may improve not only the quality of life but also the quality of sexual function [3,58,59,60]. However, there is a potential for an increased risk of local recurrence after organ-sparing surgery compared with the amputation of the penis [61]. Thus, accurate preoperative staging, together with the precise outlining of the tumor extension, infiltration depth, and tumor satellites, is important before selecting a surgical method, particularly prior to organ-sparing surgery. 

Published evidence suggests that tumors restricted to the corpus spongiosum have a better prognosis than those that invade the corpus cavernosum [55]. However, this view has been challenged in some studies that found a prognostic overlap between T2 and T3 patients or no significant differences in lymph node status in these patients [55,62,63,64,65,66]. Furthermore, some of these studies have proposed lymphovascular invasion as a significant separator of both stages [63,66]. Although tumor extent and infiltration depth are not relevant to TNM staging, it has been shown that tumors invading superficially in the corpus spongiosum (no more than 5 mm) rarely metastasize, whereas tumors that are deeply invasive in the corpus spongiosum have a higher rate of metastasis (Figure 3) [55,67].

From the penis, the tumor disseminates to the inguinal superficial lymph nodes and then to the deep and pelvic lymph nodes [55]. Skip metastasis from the superficial or deep inguinal nodes to the pelvic or other retroperitoneal nodes is extremely unusual [55]. Nodal involvement is one of the best predictors of penile cancer outcomes [31,68,69,70]. Major changes in N-staging in the eighth edition of the TNM classification include an increased number of inguinal lymph node metastases in N1 and N2 disease (Table 3).

### 3.3. Role of MRI in Staging of Penile Cancer

Current EAU-ASCO guidelines for the diagnosis and staging of penile cancer recommend the use of MRI in cases intended for organ-sparing surgery or when an invasion of the corpus cavernosum is suspected [6]. However, this recommendation is rated as weak, and the supporting evidence is sparse and limited to a few older reports [3,13]. According to these guidelines, physical examination is a reliable method for estimating penile tumor size and stage, and MRI does not outperform physical examination, at least in the ability to differentiate between T-stages T1 and T2 [6]. This rationale is supported by an older study with anticipated low-resolution images and principal technical differences from the modern mpMRI protocols currently used [71]. In general, there are some important pitfalls of physical examination that can be easily avoided using mpMRI. It is difficult to precisely estimate tumor infiltration depth through clinical examination alone; however, a strong and significant correlation between mpMRI and histopathology has been shown for both tumor size and infiltration depth [4]. Furthermore, we experienced that mpMRI can precisely identify small, deeply located tumor satellites that are difficult to detect through physical examination alone (Figure 4). The identification of tumor satellites and accurate delineation of the proximal tumor extent are important for surgical planning. Superimposed infections and body habitus can also make physical examinations difficult [31]. 

A recently published systematic review and meta-analysis showed that MRI has 86% sensitivity and 89% specificity for identifying T1 versus T2 disease, 80% and 96% for identifying T3 disease, and 86% and 93% for identifying urethral involvement [28]. The data were extracted from eight studies involving 481 patients [3,13,29,30,31,32,71,72]. The performances of MRI with and without artificial erection were comparable [28]. A recent study comparing high-resolution mpMRI and histopathological findings in the assessment of penile cancer showed very good agreement between mpMRI and histopathology for identifying T1 versus T2 disease (κ = 0.834) (Figure 3 and Figure 5), good agreement for identifying T2 versus T3 disease (κ = 0.702) (Figure 3 and Figure 4), and good agreement for identifying urethral involvement (κ = 0.746) [4]. Urethral involvement is no longer relevant for local staging; however, it is still important for surgical planning. 

The cost-effectiveness of penile mpMRI is difficult to assess due to the scarce data available. However, the incidence of penile cancer is low and there are several potential benefits arising from MRI-based assessment, such as the precise staging and delineation of the tumor extent, resulting in proper preoperative patient selection, particularly for organ-sparing procedures, with subsequent improving quality of life and sexual function and a potentially reduced number of local recurrences after surgery. 

Penile Doppler ultrasound is an alternative imaging modality recommended by EAU-ASCO for the assessment of primary tumors [6]. Bozzini et al. reported that penile ultrasound has higher sensitivity for detecting corpus cavernosum invasion than MRI with artificial erection [32]. However, the MRI protocols used in their study implemented only morphological sequences, and the technical parameters were unknown. Other studies have proven that penile ultrasound is unreliable, especially in the presence of microscopic invasion, and has important limitations, such as operator dependency and practical difficulty in assessing ulcerative tumors [3,13,73]. Several patients may also experience embarrassment and discomfort during the procedure. 

A few studies have investigated the value of MRI in assessing the N-stage in penile cancer [4,31,74]. Currently, a physical examination of both groins is recommended by the EAU-ASCO, with further diagnostic imaging for patients with clinically node-positive disease [6]. It is known that physical examination can suffer from significant false-positive and false-negative rates [31,75]. The palpation of the lymph nodes in the groins can be influenced by a simultaneous infection of the penis, where differentiation between reactive and metastatic lymph nodes is not clinically possible [31]. On the other hand, palpation can be difficult in obese or postsurgical patients and can lead to possible failure in the detection of suspicious lymph nodes [31]. In a study by Lucchesi et al., MRI performed significantly better than palpation for nodal staging [31]. Moreover, a recently published study by Barua et al. reported that the ADC derived from DWI can help to predict lymph node metastasis in penile cancer with a clinically normal groin [74]. This study reported 87% sensitivity and 81% specificity for predicting lymph node metastases using DWI [74]. The data are limited, but even with functional sequences, the detection of small lymph node metastases or small tumor amounts within the lymph nodes can be challenging using MRI [4,76]. Higher sensitivity for the detection of lymph node involvement can be achieved using fluorine-18 fluorodeoxyglucose positron emission tomography with computed tomography (18F-FDG-PET/CT) [77]. 

## 4. MRI for the Postoperative Assessment of Penile Cancer and Treatment Response

Total penectomy with perineal urethrostomy is the traditional surgical treatment for patients with penile cancer [78]. However, in recent years, organ-sparing surgery has become more common, as it can provide oncological outcomes comparable to conventional techniques in properly selected patients [78]. Organ-sparing surgery is defined as an organ-sparing complete excision of the primary tumor, with or without reconstruction, aimed at preserving sexual function and improving cosmetic outcome [78]. Several organ-sparing treatment techniques are available, including laser ablation therapy, circumcision, wide local excision, glans resurfacing, and glansectomy. Currently, controversy exists as to whether partial penectomy should be viewed as an organ-sparing procedure, considering organ-sparing partial penectomy to involve glansectomy with the shaving of the distal tips of the corpora [78]. As discussed previously, mpMRI may play an important role in the preoperative selection of patients and the determination of the most appropriate treatment options. For regional lymph node management, inguinal lymph node dissection (ILND) remains the standard of care [6]. Pelvic LND is recommended in patients with three or more positive inguinal nodes on the ipsilateral site or in the presence of extranodal extension [6]. 

From an oncological perspective, surveillance is important because the early detection of recurrence may increase the likelihood of curative treatment [6]. A recently published study of 551 patients with penile cancer who underwent ILND reported that most distant, inguinal, and pelvic recurrences occurred within 48 months of ILND, whether the majority of local recurrences occurred on a lengthy timeline, within 152 months of ILND [79]. These findings suggest the necessity of long-term follow-ups of the penis, perineum, and scrotum, whether shorter follow-ups can be suggested for distant, inguinal, and pelvic recurrences, as 95% of these occurred within 48 months of ILND [79]. 

According to clinical guidelines, a regular physician or self-examination is recommended for local surveillance after surgery [6]. However, palpation is challenging in obese patients, in patients with a short penile stump, or when the recurrences are small and located in areas affected by postoperative scar formation and fibrosis. The data are lacking; however, mpMRI may allow for the precise detection of local tumor recurrence and biopsy guidance. In our experience, functional MRI sequences can detect even small areas with increased cell density (DWI) and neoangiogenesis (DCE-MRI), and provide accurate anatomical information about the recurrent tumor extent and structures involved, which can be difficult to obtain clinically (Figure 6).

For inguinal, pelvic, and distant surveillance, whole-body cross-sectional imaging, such as 18F-FDG-PET/CT or CT, is recommended for pN+ patients [6]. In some cases, MRI with functional sequences can aid in the detection of inguinal recurrences after ILND, especially in areas affected by extensive scar formation and fibrosis (Figure 6).

Furthermore, mpMRI can be useful for the evaluation of treatment response in patients undergoing oncological therapies. MpMRI can detect morphological changes in responding tumors, such as shrinkage and necrosis, as well as functional changes, such as decreased cell density on DWI, decreased angiogenesis and vascularity, and changes in the enhancement pattern of tumor tissue on DCE-MRI, following treatment (Figure 7). 

## 5. Other Malignant and Benign Penile Lesions

### 5.1. Premalignant and Malignant Penile Lesions

Several premalignant genital lesions have been described and include penile intraepithelial neoplasia (PeIN), giant condyloma acuminata, pseudoepitheliomatous keratotic and micaceous balanitis, male lichen sclerosus, and penile cutaneous horn [80,81]. It can be challenging to provide an exact diagnosis using MRI; however, most of these lesions will not display the appearance of high-grade malignant tumors on functional sequences. Differentiated PeIN is associated with chronic inflammation and is typically not associated with HPV, whereas the undifferentiated type is generally associated with HPV [80,82]. Giant condyloma acuminata (Bushke–Lowenstein tumor) is a rare, benign condition that is typically associated with HPV positivity, and transformation to squamous cell carcinoma may occur in more than half of cases [83]. Pseudoepitheliomatous keratotic and micaceous balanitis is an uncommon disease that may have locally invasive or aggressive tendencies and may progress to verrucous or squamous cell carcinoma [84] (Figure 8). 

Squamous cell carcinoma accounts for most cases of malignant penile tumors, and other tumors include metastases, urethral cancer, malignant melanoma, basal cell carcinoma, extramammary Paget’s disease, and sarcoma [81]. 

Metastatic tumors secondarily involving the penis are uncommon. The most frequent primaries originate from the genitourinary and gastrointestinal tracts, including the prostate, urinary bladder, and colon/rectum [85] (Figure 9). The most common penile locations include the glans and corpus cavernosum [85]. It is important to carefully inspect the penis and genital region on any oncological pelvic MRI because small, metastatic lesions can be easily overlooked. 

Primary urethral cancer is rare and even more uncommon in the anterior part of the urethra [86,87]. The three main histological types are squamous cell carcinoma, followed by urothelial carcinoma and adenocarcinoma [86]. Penile MRI can aid in assessing local tumor extension and the involvement of adjacent anatomical structures (Figure 10). 

Malignant melanoma of the penis is an extremely rare malignancy that accounts for less than 2% of all primary penile malignant lesions [88]. The tumor is often located on the glans, prepuce, penile shaft, or distal urethra [88,89]. Histopathological examination provides the final diagnosis; however, melanomas can have a distinct appearance on MRI, which can help narrow down the differential diagnoses. Owing to the paramagnetic properties of melanin and blood products, malignant melanomas can appear hyperintense on T1-weighted images and hypointense on T2-weighted images [90] (Figure 11). 

### 5.2. Benign Penile Lesions

A wide range of infectious dermatoses can affect the penis, including viral, mycotic, parasitic, and bacterial infections, with the most prevalent causative agents being HPV, herpes simplex virus (HSV), candidiasis, and chlamydia [80]. 

The prevalence of subclinical or latent HPV infection can be as high as 30–50% [80,91]. Genital warts are commonly produced by HPV serotypes 6 and 11, which can further induce condyloma [80]. Although condylomas are considered to be caused by HPV subtypes distinct from those that lead to squamous cell carcinoma, infections with multiple HPV types are common, and instances of coinfection with oncogenic HPV and the malignant transformation of condyloma have been described [92] (Figure 12). 

Penile abscess is an uncommon urological condition that can be associated with penile trauma, disseminated infection, or underlying diseases, such as diabetes or malignancy [93]. Abscesses have a typical appearance on mpMRI, which can confirm the clinical diagnosis (Figure 13). A biopsy should be performed in uncertain cases to exclude underlying tumors. 

The most common benign soft-tissue tumors that affect the penis are vascular neoplasms, followed by tumors of neural, myoid, and fibrous origins [94]. Urethral cavernous hemangioma is an unusual benign tumor with few cases reported [95]. It can be easily misdiagnosed clinically; however, a quite specific MRI appearance of hemangioma has been described, which can aid in limiting differential diagnoses [95]. Gross hematuria is the most common clinical symptom and is often aggravated after erection [95]. 

## 6. Conclusions and Future Directions

MpMRI is the reference standard for oncological imaging. Although the data are scarce, mpMRI complements physical examination and plays several important roles in assessing penile cancer, including pretreatment staging, the preoperative selection of patients, postoperative local and regional surveillance, and the assessment of treatment response after local or systemic therapies. The development of functional sequences and modern mpMRI protocols allows for noninvasive examinations and the precise outlining of the tumor, which is crucial for decision making and selection of the most appropriate surgical technique from various options available. Finally, a range of lesions can arise in the male genital area, many of which have distinct MRI appearances. Thus, mpMRI can narrow down the differential diagnoses and potentially aid in differentiating between benign and malignant conditions. MpMRI should also be considered in patients with negative penile biopsies and a high clinical suspicion of cancer, as false-negative biopsies can occasionally occur. 

The development of functional sequences enables noninvasive assessment of tumor aggressiveness, and the data extracted from these images can be further linked to genotypic and phenotypic tumor patterns. For instance, the correlation between quantitative MRI parameters and immunohistochemical expression patterns of p16 in penile cancer could be important for selecting patients with a potentially poorer prognosis and reduced treatment response. Moreover, functional data extracted from mpMRI can be linked to metabolic parameters extracted from PET, allowing for the simultaneous evaluation of factors that may influence the tumor microenvironment, such as hypoxia. The impact of observer variability and the need for education, specialized training, and expertise in interpreting and reporting penile mpMRI is also not known and needs to be corroborated in future studies. In conclusion, non-erectile mpMRI is a promising imaging method that is noninvasive and allows for a comprehensive evaluation of morphological and functional alterations in tumors; however, there is a need for agreement on the minimum technical requirements for MRI protocols and prospective studies that will finally help to establish the role of mpMRI in assessing penile cancer.

## Figures and Tables

**Figure 1 cancers-15-05324-f001:**
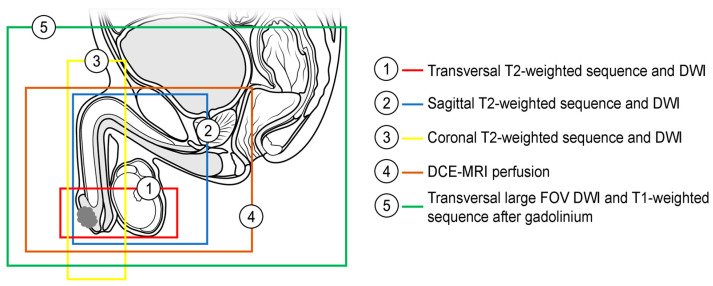
Recommended coverage of the anatomical area by morphological and functional sequences in penile MRI (DWI—diffusion-weighted imaging, DCE-MRI—dynamic contrast-enhanced MRI, FOV—field of view).

**Figure 2 cancers-15-05324-f002:**
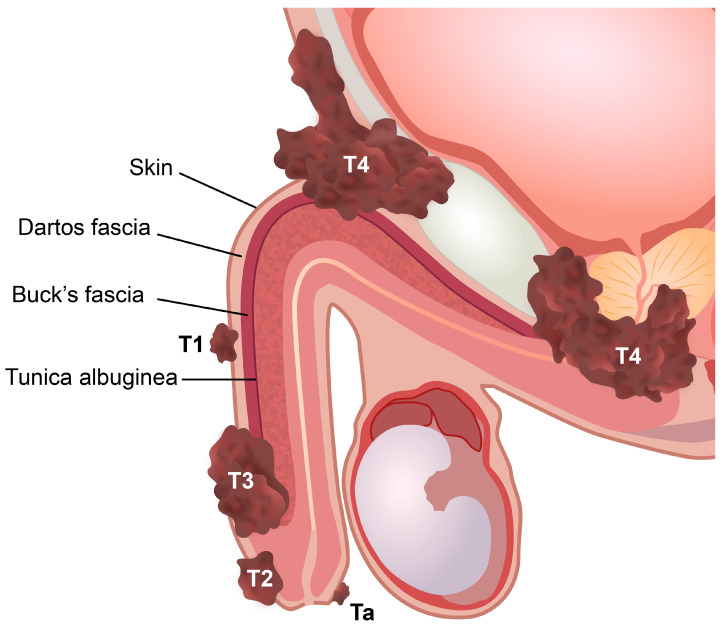
Stages of penile cancer according to the eighth edition of the Union for International Cancer Control (UICC)/American Joint Committee on Cancer (AJCC) TNM classification. Ta, noninvasive localized squamous cell carcinoma. T1, invasion of subepithelial connective tissue. T2, invasion of corpus spongiosum with or without invasion of urethra. T3, invasion of corpus cavernosum with or without invasion of urethra. T4, invasion of adjacent structures.

**Figure 3 cancers-15-05324-f003:**
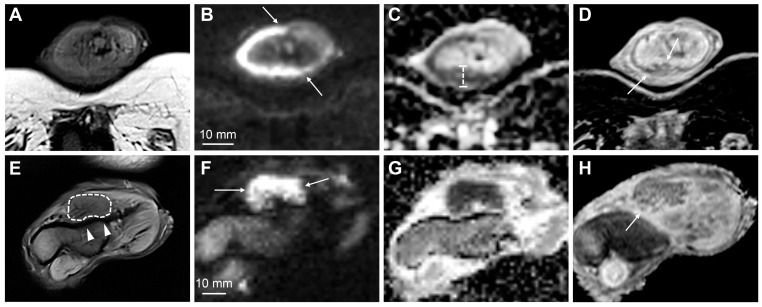
MpMRI findings of superficial (**A**–**D**) and deeply invasive (**E**–**H**) T2 penile cancer. The superficial tumor is difficult to outline on T2-weighted imaging (**A**); however, the high b-value DWI increases the conspicuity of the lesion, showing high signal intensity in tumor tissue (**B**, arrows). The tumor has a moderately low ADC (950 μmm^2^/s) (**C**). Subsequent surgery presented a good correlation between tumor thickness on mpMRI (**C**) and histopathology. The contrast-enhanced T1-weighted imaging shows tumor infiltration into the dorsal part of the glans (**D**, arrows). The lower row (**E**–**H**) shows a deeply invasive tumor in the glans (**E**—T2-weighted imaging, dashed line), located close but not infiltrating into the adjacent tunica albuginea (**E**, arrowheads). The conspicuity of the tumor and its boundaries increase on high b-value DWI (**F**, arrows), ADC (**G**) and contrast-enhanced T1-weighted imaging (**H**, arrow).

**Figure 4 cancers-15-05324-f004:**
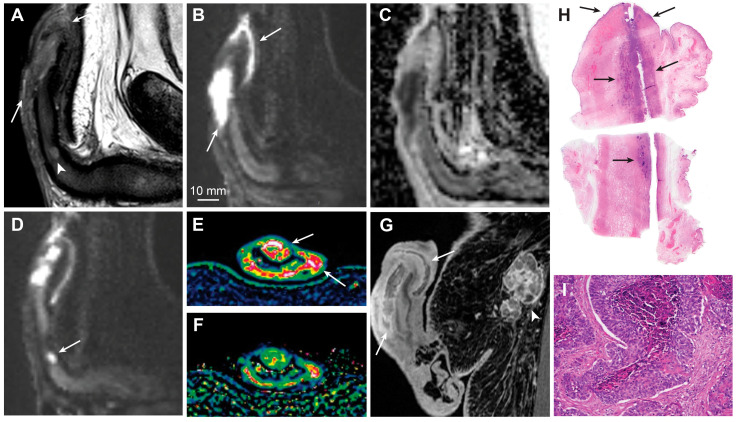
Sagittal T2-weighted imaging (**A**), high b-value DWI (**B**) and ADC (**C**) showing a large T3 penile cancer (arrows). The tumor infiltrates the corpus spongiosum and cavernosum and there is massive infiltration in the penile urethra. The small, deeply located tumor satellite in the corpus cavernosum is shown (**A**, arrowhead). This finding is crucial for surgery planning since the satellite defines the proximal tumor extent and is almost impossible to detect clinically because of its small size. The tumor satellite has a high signal intensity in DWI (**D**, arrow). The tumor has high vascular permeability, as shown on the wash-in (*K^trans^*) (**E**, arrows) and washout (*k_ep_*) (**F**) perfusion maps and contrast-enhanced T1-weighted imaging (**G**, arrows). A large inguinal lymph node metastasis is also shown (**G**, arrowhead). Photomicrograph of a whole mount hematoxylin and eosin-stained section shows a large tumor in the corpus spongiosum, cavernosum, and urethra (**H**, arrows). Photomicrograph of histologic specimen shows infiltration of basaloid squamous cell carcinoma (**I**, magnification ×10).

**Figure 5 cancers-15-05324-f005:**
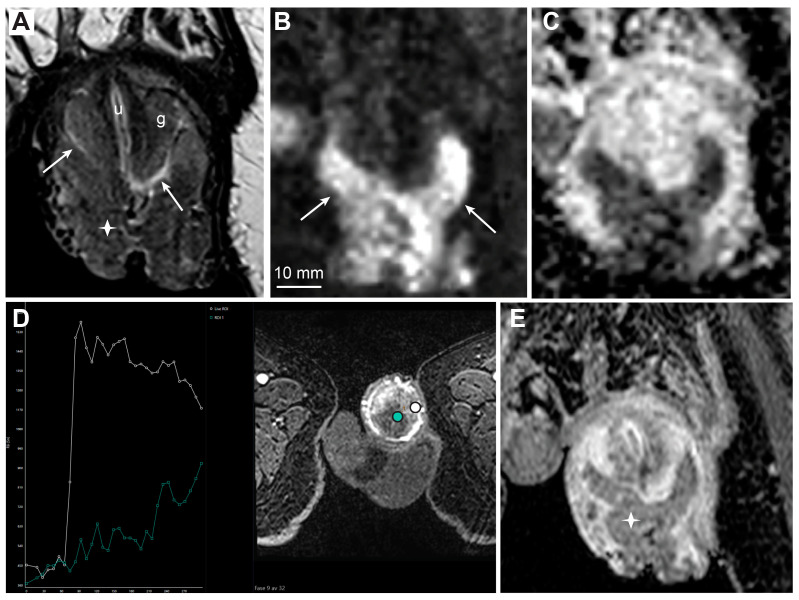
MRI findings of large T1 penile cancer. T2-weighted imaging (**A**, asterisk) and high b-value DWI (**B**, arrows) show a large prepuce tumor. The tumor lies close to but does not infiltrate the glans, with a visible fluid layer between the glans surface and tumor (**A**, arrows, g—glans, u—urethra). The ADC map displays low diffusion in the tumor (**C**). DCE-MRI (**D**) shows high permeability in the tumor with subsequent washout (**D**, white curve), in contrast to homogenously enhancing the glans with lower vascular permeability and no washout (**D**, green curve). The contrast-enhanced T1-weighted imaging shows reduced contrast enhancement in the tumor compared to surrounding structures, due to washout (**E**, asterisk).

**Figure 6 cancers-15-05324-f006:**
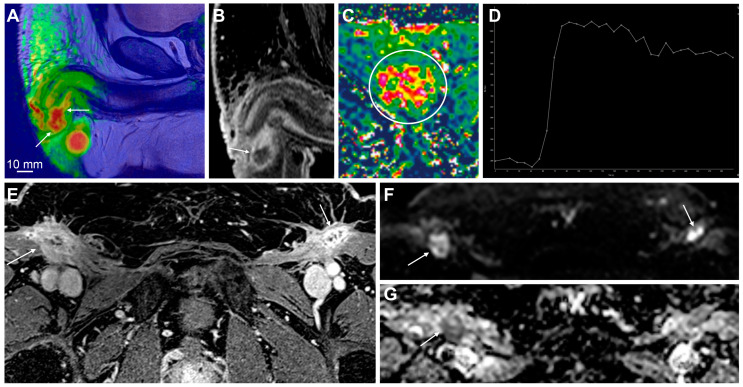
MpMRI findings of local and regional nodal recurrence. A fusion overlay of high b-value DWI and T2-weighted imaging shows local recurrence after partial penectomy (**A**, arrows). The findings are obvious in contrast-enhanced T1-weighted imaging (**B**, arrow) and DCE-MRI (**C**,**D**). There is rapid enhancement on the wash-in (*K^trans^*) perfusion map within the tumor (**C**, circle) and some washout on the perfusion curve (**D**). T1-weighted imaging shows extensive contrast enhancement in the postsurgical bed after ILND, more than can be expected from postsurgical scarring and fibrosis (**E**, arrows). DWI and ADC map show suspicious low diffusion within the enhancing tissue, consistent with tumor infiltration (**F**,**G**, arrows). Histopathological examination confirmed both local and regional nodal recurrence from penile cancer.

**Figure 7 cancers-15-05324-f007:**
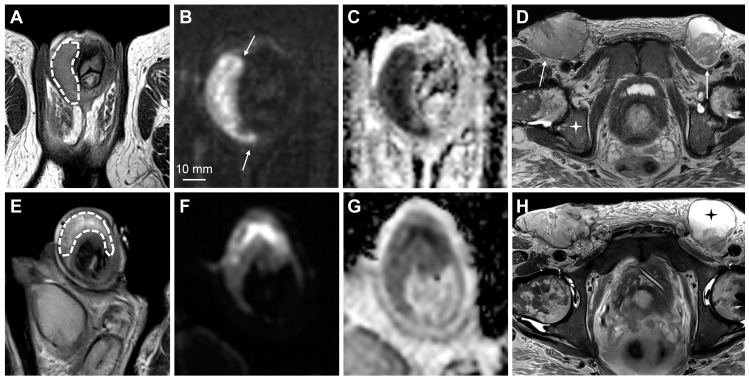
Evaluation of the treatment response on penile MRI. The upper row (**A**–**D**) shows pretreatment findings of large penile cancer on T2-weighted imaging (**A**, dashed line), high b-value DWI (**B**, arrows), and ADC map (**C**). Large inguinal lymph node metastases are also shown (**D**—T2-weighted imaging, arrows). Note the pathological signal in the bone marrow, secondary to known chronic myelogenous leukemia (**D**, asterisk). The lower row (**E**–**H**) shows the posttreatment findings after the start of chemotherapy. The tumor has a stable volume; however, high signal intensity on T2-weighted imaging has appeared since the pretreatment scan, consistent with necrosis and the treatment response (**E**, dashed line). The ADC is slightly higher, which is also consistent with the response (**G**). There is some treatment response in the inguinal lymph nodes with the slight shrinkage of solid tumor tissue and increased necrosis on T2-weighted imaging (**H**, asterisk).

**Figure 8 cancers-15-05324-f008:**
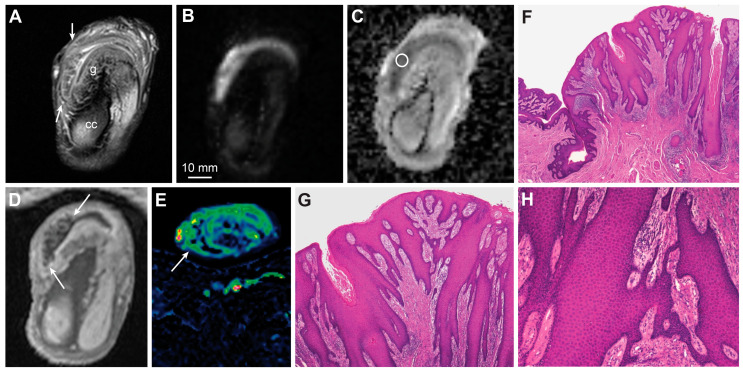
MRI findings of pseudoepitheliomatous hyperplasia with inflammation. T2-weighted imaging shows an exophytic lesion along the inner layer of the prepuce (**A**, arrows, g—glans, cc—corpus cavernosum). High b-value DWI (**B**) and ADC map (**C**, circle) show heterogenous yet predominantly high diffusion in the lesion (ADC > 1300 μmm^2^/s). Contrast-enhanced T1-weighted imaging displays an exophytic lesion with a papillomatous surface along the inner prepuce and glans (**D**, arrows). The lesion is vascularized; however, the permeability rate is very low, as shown on the wash-in (*K^trans^*) perfusion map (**E**, arrow). Photomicrographs of histologic specimens display pseudoepitheliomatous hyperplasia with inflammation (**F**—magnification × 2, **G**—magnification × 4, **H**—magnification × 10).

**Figure 9 cancers-15-05324-f009:**
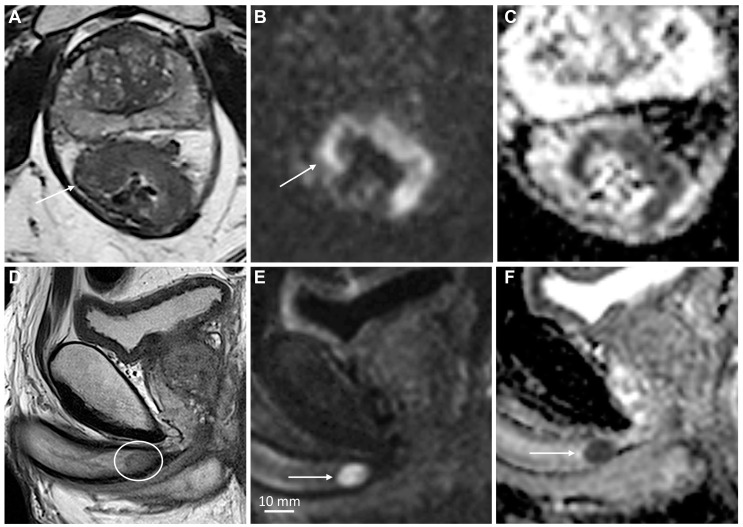
MRI findings of penile metastasis originating from rectal cancer. The upper row shows low, locally advanced rectal cancer (arrows) (**A**—T2-weighted imaging, **B**—DWI, **C**—ADC). The lower row (**D**–**F**) shows penile metastasis located in the posterior part of the left corpus cavernosum. The lesion can be easily overlooked on T2-weighted imaging (**D**, circle); however, high b-value DWI (**E**, arrow) and ADC (**F**, arrow) increase the conspicuity of the tumor.

**Figure 10 cancers-15-05324-f010:**
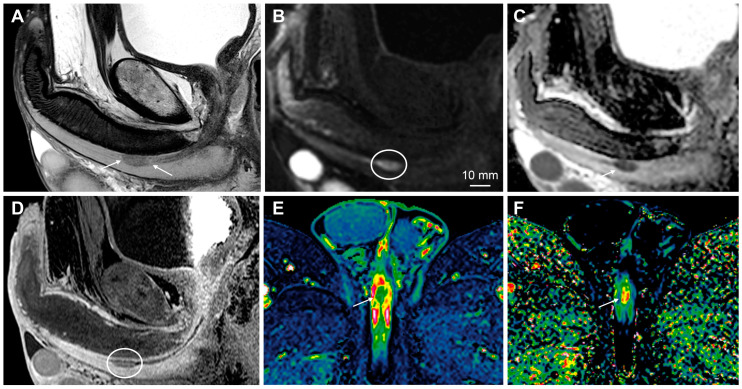
MpMRI findings of urethral cancer. T2-weighted imaging (**A**, arrows) and high b-value DWI (**B**, circle) show a small lesion along the spongy urethra. The lesion has low diffusion on the ADC map (**C**, arrow), and displays an enhancement pattern typical for malignant tumors in contrast-enhanced T1-weighted imaging (**D**, circle) and DCE-MRI (**E**,**F**, arrows). The wash-in (*K^trans^*) perfusion map shows early enhancement in the lesion, but in a lesser grade than the adjacent, rich vascularized corpus spongiosum (**E**, arrow). The *k_ep_* perfusion map displays subsequent washout in the tumor (**F**, arrow). The histological evaluation confirmed urethral squamous cell carcinoma.

**Figure 11 cancers-15-05324-f011:**
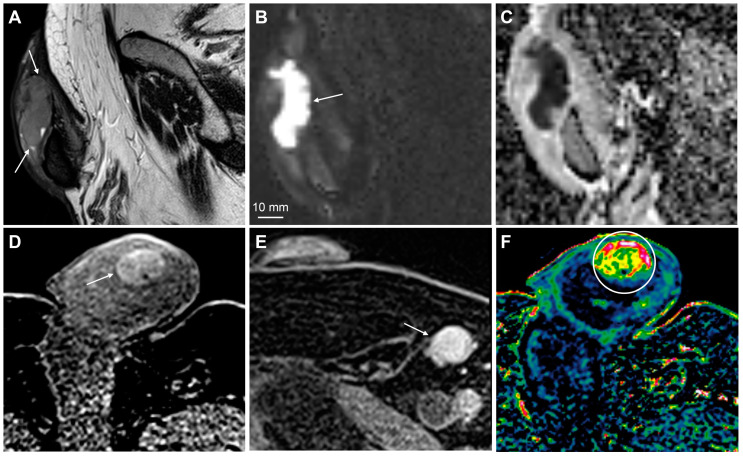
MpMRI findings of penile malignant melanoma. A large, cell-dense tumor in the navicular fossa of spongy urethra is detected in T2-weighted imaging (**A**, arrows), high b-value DWI (**B**, arrow), and ADC (**C**). The primary tumor (**D**, arrow) and inguinal nodal metastasis (**E**, arrow) are hyperdense on T1-weighted imaging because of the paramagnetic effect of melanin and/or blood products. The tumor is richly vascularized, as shown in the wash-in (*K^trans^*) perfusion map (**F**, circle).

**Figure 12 cancers-15-05324-f012:**
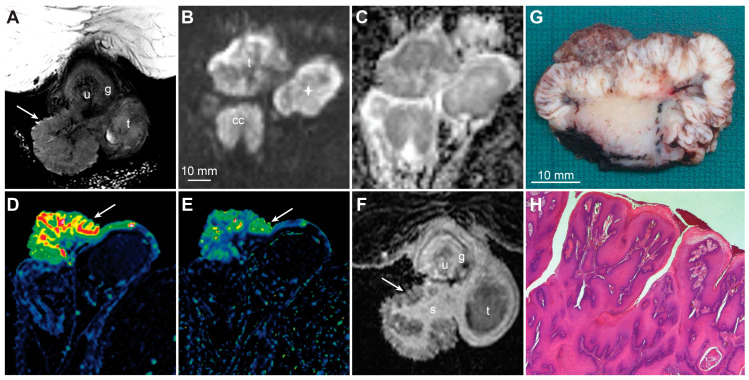
T2-weighted imaging showing a large exophytic tumor located just inferior to the glans and consistent with a condyloma (**A**, arrow, g—glans, u—urethra, t—testicle). The high b-value DWI (**B**, t—tumor, cc—corpus cavernosum, asterisk—testicle) and ADC (**C**) display predominantly high diffusion in the tumor (ADC > 1000 μmm^2^/s). The tumor has a cauliflower-like appearance in DCE-MRI (**D**—wash-in (*K^trans^*) map, **E**—washout (*k_ep_*) map) and contrast-enhanced T1-weighted imaging (**F**). The diffusion is high; however, the papillomatous surface shows a high vascular permeability, with subsequent washout (**D**–**F**, arrows, s—tumor stalk, u—urethra, g—glans, t—testicle). A photomacrograph of formalin-fixed specimen shows a large, cauliflower-like tumor, consistent with a condyloma (**G**). Photomicrograph of histologic specimen shows condyloma with development of well-differentiated squamous cell carcinoma (**H**, magnification × 2).

**Figure 13 cancers-15-05324-f013:**
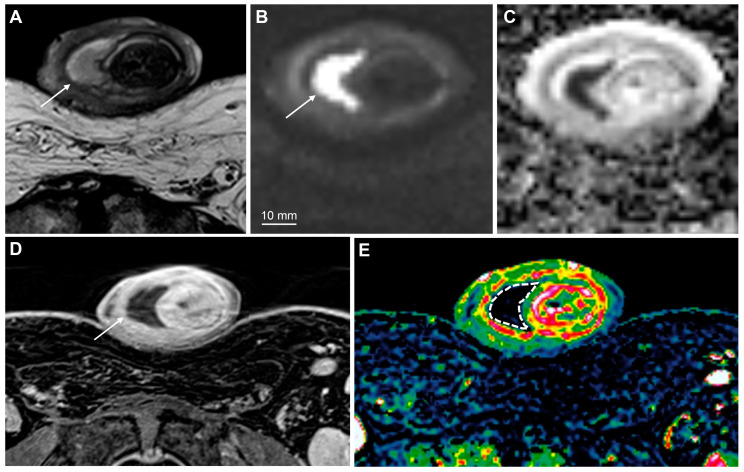
T2-weighted imaging (**A**, arrow) and high b-value DWI (**B**, arrow) show a well-demarcated lesion within the right part of the glans. The lesion has very low diffusion, and the ADC is lower than that expected from the tumor (ADC = 300 μmm^2^/s) (**C**). The lesion is not vascularized, as shown in the contrast-enhanced T1-weighted imaging (**D**, arrow) and wash-in (*K^trans^*) perfusion map (**E**, dashed line). The findings are consistent with the penile abscess. The biopsy was performed to exclude underlying malignancy but was negative.

**Table 1 cancers-15-05324-t001:** MpMRI protocol for the assessment of primary penile carcinoma [4].

The examination is performed without artificial erection; Penis in anatomical position, pointing downward along the midline; Small FOV, high-resolution T2-weighted sequence in three anatomical planes (transversal, coronal and sagittal) over primary tumor. Sagittal T2-weighted sequence should cover the whole urethra; Small FOV, high-resolution DWI in three anatomical planes (transversal, coronal and sagittal) over primary tumor. Sagittal DWI should cover the whole urethra; Transversal large FOV DWI covering inguinal and pelvic lymph nodes; Transversal DCE-MRI perfusion over penis and urethra; Transversal high-resolution 3D GE T1-weighted Dixon sequence after gadolinium over penis, urethra, and inguinal and pelvic lymph nodes.

DWI—diffusion-weighted imaging; DCE-MRI—dynamic contrast-enhanced magnetic resonance imaging; FOV—field of view; GE—gradient echo.

**Table 2 cancers-15-05324-t002:** The World Health Organization (WHO) 2022 classification of invasive penile carcinomas. Adapted with permission from [1].

HPV-associated squamous cell carcinoma	Basaloid Warty Clear cell Lymphoepithelioma-like Mixed
HPV-independent squamous cell carcinoma	Usual type (includes pseudohyperplastic and pseudoglandular) Verrucous carcinoma (includes carcinoma cuniculatum) Papillary Sarcomatoid Mixed
Squamous cell carcinoma; NOS	
Adenosquamous carcinoma (includes mucoepidermoid carcinoma)	

HPV—human papillomavirus; NOS—not otherwise specified.

**Table 3 cancers-15-05324-t003:** The eighth edition of Union for International Cancer Control (UICC)/American Joint Committee on Cancer (AJCC) TNM classification for the pathological staging of penile cancer. Adapted with permission from [56].

pT-Primary Tumor	TX	Primary tumor cannot be assessed
	T0	No evidence of primary tumor
	Tis	PeIN
	Ta	Noninvasive localized squamous cell carcinoma
	T1	Invasion of subepithelial connective tissue: T1a—no lymphovascular invasion or perineural invasion in not poorly differentiated tumor T1b—lymphovascular invasion and/or perineural invasion or poorly differentiated tumor
	T2	Invasion of corpus spongiosum with or without invasion of urethra
	T3	Invasion of corpus cavernosum with or without invasion of urethra
	T4	Invasion of adjacent structures
pN-Regional Lymph Nodes	pNX	Regional lymph nodes cannot be assessed
	pN0	No regional lymph node metastases
	pN1	Metastasis in ≤ 2 unilateral inguinal lymph nodes
	pN2	Metastasis in ≥ 3 unilateral inguinal lymph nodes or bilateral inguinal lymph nodes
	pN3	Metastasis in pelvic lymph node(s) or ENE
pM-Distant Metastasis	pM0	No distant metastasis
	pM1	Distant metastasis microscopically confirmed

PeIN—penile intraepithelial neoplasia; ENE—extranodal extension.

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
