# Peer review of "Multiparametric Magnetic Resonance Imaging of Penile Cancer: A Pictorial Review"

_cancers, 2023, doi:10.3390/cancers15225324_

Round 1

Reviewer 1 Report

Comments and Suggestions for Authors

The manuscript titled “Multiparametric Magnetic Resonance Imaging of Penile Cancer: A Pictorial Review” by Switlyk, M.D.; et al. is a Review work where the authors outline the state-of-the-art and the most recent advances according to the use of multiparametric magnetic resonance imaging (mpMRI) in the detection of penile cancer malignancies. The authors show some guidelines of how mpMRI could aid to the accurate prognosis of this disease that significantly serve in the most convenient decision-making to choose the right treatment. This review work could be interesting for a certain target audience and its content is well-organized.

However, it exists some points that need to be addressed (please, see them below detailed point-by-point). The most relevant outcomes found by the authors can contribute in the growth of many fields like the linked to the healthcare, overall focused in the diagnose and treatment of human tumoral diseases. For this reason, I will recommend the present scientific manuscript for further publication in Cancers once all the below described suggestions will be properly fixed.

Here, there exists some points that must be covered in order to improve the scientific quality of the manuscript paper:

1) KEYWORDS (OPTIONAL). The authors should consider to add the full name of the terms “DCE” and “DWI” instead of their respective abbreviated forms.

2) INTRODUCTION. This section is clear and concise. “Primary penile cancer is an uncommom malignancy with an incidence (…) than 1 per 100,000 in North America and Europe” (lines 41-43). Could the authors quantify the most relevant incidence of penile cancer disease in terms of age and other relevant risk factors? [1].

[1] Cardona, C.E.M.; García-Perdomo, H.A. Incidende of penile cancer worldwide: systematic review and meta-analysis. Rev. Panam. Salud Publica 2017, 41, e117. https://doi.org/10.26633/RPSP.2017.117.

3) MRI IN THE ASSESSMENT OF PENILE CANCER. The authors should add a schematic representation depicting the working principles of MRI. This information will benefit the potential readers and other stakeholders to better understand this technique.

4) “It involves sequentially acquired T1-weighted images collected before, (…) gadolinium-based contrast agent” (lines 89-91). Firstly, why gadolinium nanoparticles display a better performance for MRI compared to other magnetic nananoparticles? (e.g. strong positive contrast enhancement properties, high magnetic moment, high-biocompatibility and stability, …). A brief statement should be provided in this regard. Furthemore, during the main manuscript body text the authors deeply explain the MRI technique, but it lacks to list other bulk techniques like magneto-optical Kerr effect [2] or Mössbauer spectroscopy [3] and single molecule approaches [4] capable to measure magnetic signals and the advantages treasured of MRI respect to this technique (e.g. non-invasive technique, versatility or the possibility to carry out longitudinal studies to track the patient progress over time, among others).

[2] Yamamoto, S.; Matsuda, I. Measurement of the Resonant Magneto-Optical Kerr Effect Using a Free Electron Laser. Appl. Sci. 2017, 7, 662. https://doi.org/10.3390/app7070662.

[3] Kuzmann, E.; Homonnay, Z.; Klencsár, Z.; Szalay, R. 57Fe Mössbauer Spectroscopy as a Tool for Study of Spin States and Magnetic Interactions in Inorganic Chemistry. Molecules 2021, 26, 1062. https://doi.org/10.3390/molecules26041062.

[4] Winkler, R.; Ciria, M.; Ahmad, M.; Plank, H.; Marcuello, C. A Review of the Current State of Magnetic Force Microscopy to Unravel the Magnetic Properties of Nanomaterials Applied in Biological Systems and Future Direction for Quantum Technologies. Nanomaterials 2023, 13, 2585. https://doi.org/10.3390/nano13182585.

5) “We perform the examinations (…) high image resolution and signal-to-noise ratio (SNR) within a reasonable scan time” (lines 169-170). The authors need to quantify these three parameters (image resolution, signal-to-noise- ratio and the employed scan time).

6) Finally, the authors should also describe the alternative employed diagnostic methods for penile cancer as ultrasonography examination or incisional biopsy (this latest case is partially already discussed by the authors in the lines 396-401).

7) CLASSIFICATION AND STAGING OF PENILE CANCER. Figure 2 (line 276). The authors should add the lateral scale bar in the images to compare the size of the tumour between all the tested conditions. Same comment for Figure 3 (line 307), Figure 4 (line 333), Figure 5 (line 406), Figure 6 (line 427), Figure 7 (line 454) Figure 8 (line 457), Figure 9 (line 486), Figure 10 (line 507), Figure 11 (line 530), and Figure 12 (line 554).

8) MRI FOR THE POSTOPERATIVE ASSESSMENT OF PENILE CANCER AND TREATMENT RESPONSE and OTHER MALIGNANT AND BENIGN PENILE LESIONS (Sections 4 and 5, respectively). The information provided in these section is accurate. No actions are requested from the authors.

9) CONCLUSIONS AND FUTURE DIRECTIONS. Here, the authors summarize and remark the most relevant outcomes shown in this Review work.  The authors should consider also to mention the potential applications of mpMRI not only for penile cancer, but also for other malignancies like prostate cancer or bone sarcomas, among others.

10) REFERENCES. The references are mostly in the proper format style of Cancers. The authors need to check the guidelines concerning this point.

Comments on the Quality of English Language

The authors should recheck the English out before to send the manuscript revised version in order to fix some aspects susceptible to be improved.

Reviewer 2 Report

Comments and Suggestions for Authors

Excellent manuscript and timely review of current standards in MRI imaging for patients with penile cancer, the authors demonstrate a clear understanding of the various diagnostic and therapeutic paradigms and highlight how MRI can support clinical decisions and survivorship of patients with this rare disease.

Reviewer 3 Report

Comments and Suggestions for Authors

The article discusses the role of multiparametric magnetic resonance imaging (mpMRI) in assessing penile cancer. Accurate preoperative staging and precise outlining of the tumor extent are crucial for selecting the most suitable treatment approach and improving outcomes. The current clinical staging of penile cancer is still primarily based on physical examination, and mpMRI is a vital imaging modality that complements physical examination and reduces uncertainties that can easily arise during this examination. The article focuses on the diagnostic performance of non-erectile mpMRI in evaluating penile cancer, reviewing the use of functional techniques for comprehensive oncological assessment, the current literature, and the latest guidelines. The article also discusses the principles of non-erectile mpMRI, including functional techniques and their applications in evaluating the male genital region, along with clinical protocols and technical considerations.

There are some limitations and shortcomings of the article which need to be addressed before acceptance:

- The article focuses only on the diagnostic performance of non-erectile mpMRI in evaluating penile cancer and does not cover other imaging modalities or techniques such as US, CEUS or CT.

- The article acknowledges that evidence for the application of MRI in assessing penile cancer is scarce, and there is no consensus on MRI protocol.

- The article also notes that functional MRI techniques have not been widely studied before, and the applications of advanced techniques in penile MRI are yet to be defined.

- The article highlights the need for prospective and feasible multicenter trials due to the rarity of the disease, which needs to be addressed in detail.

- The article does not critically analyze the studies or sources it cites or compare or contrast different studies or references.

- The article does not discuss the limitations of mpMRI in general, such as motion artifacts, susceptibility to spasms and muscle movements, and intrinsic patient characteristics that may affect image quality.

- The article needs to provide a comprehensive overview of the current state of research on the topic and discuss the limitations or shortcomings of the studies it cites.

- The article does not provide any practical recommendations or guidelines for clinicians or researchers, nor does it discuss the implications of its findings for clinical practice or future research.

- The article does not discuss the potential risks or harms associated with mpMRI, such as using contrast agents, claustrophobia, and anxiety.

- The article needs to clearly define non-erectile mpMRI or explain how it differs from other types of MRI.

- The article does not discuss the cost-effectiveness of mpMRI in the assessment of penile cancer, which may be a significant limitation in some healthcare settings.

- The article does not provide a clear conclusion or summary of its findings, nor does it discuss the implications of its results for future research or clinical practice.

- The article does not discuss the potential limitations or biases of the studies or sources it cites, nor does it provide a critical analysis of the quality of the evidence.

- The article does not discuss the potential impact of patient factors on the accuracy of mpMRI in assessing penile cancer.

- The article does not provide a detailed description of the technical considerations or requirements for performing mpMRI, which may limit its usefulness for clinicians or researchers who are not familiar with the technique

- The article needs to discuss the potential limitations or challenges of interpreting mpMRI results, such as interobserver variability and the need for specialized training and expertise.

- Illustration with stages of cancer is recommended

Overall, while the article provides a helpful overview of the diagnostic performance of non-erectile mpMRI in evaluating penile cancer, it has several limitations and shortcomings that should be considered when interpreting its findings.

Comments on the Quality of English Language

MINOR

Round 2

Reviewer 1 Report

Comments and Suggestions for Authors

The authors did a great deal of work to cover many of the reviewers' suggestions. For this reason, the scientific quality of the manuscript was greatly improved (which will benefit to the potential readers).

Nevertheless, I still consider the appropriateness to introduce the advantages offered by magnetic resonance (MRI) technique compared to other available approaches to address the magnetic response of the tested contrast agents. In this context, MRI treasures much greater detection sensitity than magneto-optical kerr effect (MOKE) [1], does not require the presence of isotopes in the sample as Mössbauer spectroscopy [2] or time-consuming calibration methodologies like magnetic force microscopy [3]. For all this, MRI is the best choice to monitor the progress not only for penile tumours but also, for other types of carcinogenic human diseases.

[1] Yamamoto, S.; Matsuda, I. Measurement of the Resonant Magneto-Optical Kerr Effect Using a Free Electron Laser. Appl. Sci20177, 662. https://doi.org/10.3390/app7070662.

[2] Kuzmann, E.; Homonnay, Z.; Klencsár, Z.; Szalay, R. 57Fe Mössbauer Spectroscopy as a Tool for Study of Spin States and Magnetic Interactions in Inorganic Chemistry. Molecules 202126, 1062. https://doi.org/10.3390/molecules26041062.

[3] Winkler, R.; Ciria, M.; Ahmad, M.; Plank, H.; Marcuello, C. A Review of the Current State of Magnetic Force Microscopy to Unravel the Magnetic Properties of Nanomaterials Applied in Biological Systems and Future Direction for Quantum Technologies. Nanomaterials 202313, 2585. https://doi.org/10.3390/nano13182585. 
